# Serum-Exosome-Derived miRNAs Serve as Promising Biomarkers for HCC Diagnosis

**DOI:** 10.3390/cancers15010205

**Published:** 2022-12-29

**Authors:** Tao Rui, Xiaobing Zhang, Jufeng Guo, Aizhai Xiang, Ning Tang, Jian Liu, Zonglei Mao

**Affiliations:** 1Department of Surgery, Affiliated Hangzhou First People’s Hospital, Zhejiang University School of Medicine, Hangzhou 310003, China; 2The Center for Integrated Oncology and Precision Medicine, Affiliated Hangzhou First People’s Hospital, Zhejiang University School of Medicine, Hangzhou 310003, China

**Keywords:** biomarker, exosome, sRNA-seq, HCC, miRNA

## Abstract

**Simple Summary:**

We aim to discover the ideal biomarkers for diagnosing HCC, by detecting the small RNAs (sRNAs) from serum exosomes. After analyzing the proportion and distribution of sRNAs species from serum exosomes, we found that miRNAs were the most important components in serum exosomes. Three serum-exosome-derived miRNAs (miR-122-5p, let-7d-5p, and miR-425-5p) had significant values for diagnosing HCC, both in the training set and validation set. Our results concluded that serum-exosome-derived miRNAs could be the promising biomarkers in HCC diagnosis

**Abstract:**

Background: Serum exosomes are emerging as key liquid biopsy biomarkers for the early diagnosis of cancer. However, the proportion and distribution of small RNA (sRNA) species from serum exosomes of hepatocellular carcinoma (HCC) patients remain unclear. Effective and reliable biomarkers for HCC diagnosis should be explored. Methods: In this study, we aimed to use sRNA sequencing to profile the sRNAs of serum exosomes in HCC and non-tumor donors. The serum exosomes of 124 HCC patients and 46 non-tumor donors were enrolled for detecting the values of the potential biomarkers for the diagnosis of HCC. Results: We found that miRNAs accounted for the maximal percentage of all types of sRNAs both in the serum exosomes of HCC patients and non-tumor donors. This indicated that the serum-exosome-derived microRNAs (miRNAs) were the most valuable as potential biomarkers in HCC diagnosis. Then, miRNAs were set as research candidates. In our Chinese cohorts, three serum-exosome-derived miRNAs (miR-122-5p, let-7d-5p, and miR-425-5p) could be promising biomarkers for distinguishing HCC patients from non-tumor donors. In addition, they were preferred for the early diagnosis of HCC. We also presented the base distribution of some novel serum-exosome-derived miRNAs and described the potential values as biomarkers. Conclusions: The results suggested that the serum-exosome-derived miRNAs were the most crucial sRNA species and they highlighted the potential of serum-exosome-derived miRNAs as promising biomarkers for HCC diagnosis.

## 1. Background

Hepatocellular carcinoma (HCC) is one of the most prevalent malignancies worldwide. Although great advances have been achieved in the therapeutic strategy, HCC-related mortality has increased. Finding effective and rapid screening tools for diagnosing HCC is essential. Serum alpha-fetoprotein (AFP) measurement is the unique marker that is broadly recommended for the screening of HCC. However, long-term data indicate that the sensitivity of AFP for hepatocellular cancer only ranges from 41% to 65% [1]. The breakthrough of serum markers for the in-clinic fast screening of HCC is limited. Thus, it is essential and urgent to discover ideal markers for the early diagnosis of HCC.

Exosome, as the kind of nanoscale vesicle produced from living cells, is a rising star in tumor liquid biopsy [2,3,4]. Exosome-based liquid biopsy is a promising technological advancement for cancer diagnosis [5,6,7]. Tumor-related serum exosomes are excreted from living tumor cells [8,9]. In addition, the serum exosomes can be acquired with less invasiveness. This means that continuous information from tumors could be acquired from serum exosomes in real time [10]. Moreover, the phospholipid bilayer membranes of exosomes contribute to the regulation of membrane dynamics for exosome release [11]. The phospholipid bilayer also prevents the degradation of exosomal contents [12]. Thus, signals emitted from tumor cells can be well stored and transferred via exosomes. 

Small RNAs (sRNAs) are one of main components that exist in the exosomes [13]. Various sRNAs have been found in human-generated exosomes, including microRNAs (miRNAs), transfer RNAs (tRNAs), ribosomal RNAs, piwi-interacting RNA (piRNAs), and other unidentified RNAs. During the last two decades, a plethora of studies have confirmed that exosome-derived sRNAs have crucial value in tumor diagnosis and prognosis prediction [14,15,16]. The components of serum exosomes, such as long noncoding RNAs (lncRNAs), circle RNAs, and miRNAs have also been shown to be potential biomarkers for the diagnosis of HCC [17]. However, the proportion and distribution of sRNAs in serum exosomes of HCC are not understood, and this is critical for exploring effective tumor biomarkers. To date, no excellent exosome-derived biomarkers for diagnosing HCC have been used in clinical applications. In this study, we showed the specificity of HCC serum-exosome-derived sRNAs species and exhibited that miRNAs were the most crucial components in serum-exosome-derived sRNAs. Additionally, we introduced the top three serum-exosome-derived miRNAs as optimal biomarkers for diagnosing HCC in the clinic. 

## 2. Materials and Methods

### 2.1. Patients and Specimens 

To avoid the influence of medical interventions on serum-exosome-derived sRNA levels, all of the samples used in this study were collected prior to the patients receiving hepatectomy, tumor invasive biopsy, or any forms of tumor therapy. After the patient was diagnosed with HCC according to histopathologic examination, the collected samples from the patients could be enrolled. Finally, the serum of 127 HCC patients and 46 non-tumor donors was enrolled. Three samples were excluded from this study (one of them was accompanied by duodenal cancer and two underwent adjuvant therapy from an outside hospital before serum collection and tissue biopsy). Thus, 124 cases of HCC serum were prepared for exosome purification. The characteristics of HCC patients are shown in Appendix A. Among them, 34 (27.4%) were female and 90 (72.6%) were male, with the mean age of 59.60 ± 8.56 years. Ninety-four (75.8%) had a positive hepatitis B virus (HBV) infection, and thirty (24.2%) were of HBV-negative status. The histological stage of the HCC patients was assessed according to the 8th Edition of American Joint Committee on Cancer; the tumor, node, metastasis classification (AJCC-TNM) [18]. Of these, 65 (52.4%) had a low AJCC stage (stage I and II) and 59 (47.6%) had a high AJCC stage (stage III and IV). This research protocol was approved by the Affiliated Hangzhou First People’s Hospital, Zhejiang University School of Medicine. The participants provided written informed consent to participate in this study.

### 2.2. Serum Exosome Purification and Exosome RNA Extraction 

As previously described [19], before exosome extraction, the serum from HCC patients and non-tumor donors was centrifuged at 3000× *g* at 4 °C for 5 min and filtered with a Millex GV filter unit (0.22 μm, Merck Millipore, Darmstadt, Germany). Serum exosome purification and exosome RNA extraction from 1 mL of serum were performed by using an exoRNeasy Midi Kit (No. 77144, Qiagen, Hilden, Germany), according to the manufacturer’s instructions.

### 2.3. sRNA-Sequencing Analysis

sRNA sequencing analysis was used to screen the potential serum-exosome sRNAs for HCC diagnosis. After the candidates are selected, qRT-PCR should be further performed in large samples to conclude that the candidates could be used as ideal biomarkers. Thus, four HCC patients and four non-tumor donors were randomly selected for the sRNAs-sequencing analysis. The characteristics of the four patients and four non-tumor donors are shown in Appendix A. The sRNA-sequencing was performed by using the Multiplex Small RNA Library Prep Set for Illumina^®^ (San Diego, CA, USA) at the Novegene Bioinformatics Technology Co., Ltd. (Beijing, China), according to the manufacturer’s instructions. For annotating exosome sRNAs, the sRNAs were filtered from clean reads according to the range of length. The sRNA tags were mapped and matched to genome version CRCh38 by the short read aligner named as Bowtie [20]. For identifying the known miRNAs, the sRNAs were mapped with miRBase20.0 [21]. For predicting novel miRNAs, the available software miREvo and mirdeep2 were integrated [22,23]. Differentially expressed miRNAs between HCC patients and non-tumor donors were performed by using the DESeq R package. The log_2_ fold change > 1 and false discovery rate (FDR) (adjusted *p* value) < 0.05 were set as the threshold for distinguishing differentially expressed miRNAs. 

### 2.4. Exosome Identification

We prepared the exosome identification by transmission electron microscope (TEM) and nanoparticle tracking analysis (NTA). As previously described [19], the purified exosomes from the serum of HCC patients and non-tumor donors were negatively stained with phosphotungstic acid (2%) on a copper grid for 1.5 min and dried for 15 min. The exosomes were observed with a TEM (Tecnai G2 Spirit 120 kV, Thermo Fisher Scientific, Waltham, MA, USA). The TEM was operated by the core facility platform of Zhejiang University. The NTA was performed with the Nano-ZS90 instrument (Malvern, UK), according to the manufacturer’s instructions, for measuring the hydrodynamic diameter distribution of exosomes.

### 2.5. Quantitative Reverse Transcription Polymerase Chain Reaction (qRT-PCR)

As previously described [24], stem-loop miRNA qRT-PCR was performed for detecting the expression of miRNAs. According to the Bulge-Loop miRNA qRT-PCR Starter Kit (C10211, Ribobio, Guangzhou, China), specific reverse transcription primers were used to synthesize cDNA. PCR amplification was performed with FS Essential DNA Green Master (4913914001, Roche, Basel, Switzerland) accompanied by the LightCycler480 instrument (Roche, Basel, Switzerland). The data from the qRT-PCR were performed in replicates of three. The primers are listed in Appendix A.

### 2.6. Western Blot Analysis

As previously described [19], the total proteins were extracted from the purified exosomes. After the proteins were denatured, they were electrophoresed on sodium dodecyl sulfate–polyacrylamide gel electrophoresis SDS-PAGE gel (FD341-100, Fudebio, Hangzhou, China) and transferred onto equilibrated polyvinylidene difluoride membranes. The membranes were then blocked for 1 h at room temperature and incubated with primary antibodies overnight at 4 °C. Before being detected by an enhanced chemiluminescence (ECL) system (Biotanon, Shanghai, China) with FDbio-FemtoECL (FD8030, Fudebio, Hangzhou, China), the membranes were incubated with a secondary anti-rabbit immunoglobulin G (IgG) horseradish peroxidase (HRP)-linked antibody. The antibodies are listed in Appendix A.

### 2.7. Statistical Analysis

Mean ± standard deviation (SD) or median ± interquartile range (IQR) was used for the description of the quantitative variables. The Student’s *t*-test or the Mann–Whitney test were performed to analyze the comparison of quantitative variables between the two groups. The analysis of variance (ANOVA) followed by Newman–Keuls individual comparisons were used among more than two groups. Categorical measures were compared by the chi-square test or Fisher’s exact test. The area under receptor operated curve (AUROC) mode and the diagnostic performance characteristics were used to calculate the diagnostic values of miR-425-5p, let-7d-5p, and miR-122-5p in HCC. Statistical analysis was performed with Statistical Product and Service Solutions (SPSS) software (version 19.0, IBM, Armonk, NY, USA), and the *p* value < 0.05 was set as the significance level. *, **, ***, and **** were used to represent *p* value of < 0.05, < 0.01, < 0.001, and < 0.0001, respectively. 

## 3. Results

### 3.1. Profiling and Analyzing of Exosome-Derived Small RNAs in the Serum of HCC Patients

First, the serum exosomes from HCC patients and non-tumor donors were isolated (Figure 1A). The TEM observed the typical membrane structure of the exosomes with a diameter of less than 100 nm (Figure 1B). The hydrodynamic diameter distribution of serum exosomes from HCC patients and non-tumor donors was shown (Appendix A). After extracting the proteins of the exosomes, we detected four classic protein markers of exosomes (CD9, CD63, TSG101, and HSP70) by Western blot analysis (Figure 1C and Appendix A). Thus, the exosomes were successfully extracted from serum.

We further prepared the sRNAs’ sequencing. The sequencing error rate of each base was less than 0.05%, which suggested that the sequencing results were of high quality (Appendix A). By analyzing the proportions of the components of sRNAs (miRNA, tRNA, rRNA, snRNA, and snoRNA) in the serum exosomes, we markedly observed that miRNAs were the kind of sRNAs with the largest numbers, compared with the others (Figure 1D). Notably, the proportions of miRNAs were greater than 50%, and 40% in the HCC-derived serum exosomes and non-tumor donor-derived serum exosomes, respectively (Figure 1D). The results suggested that miRNAs were the critical component in serum-exosome-derived sRNAs. Interestingly, the length distributions of the sRNAs exhibited high concordance. The sRNAs with a length of 19–22 nucleotides (nt) had significant peaks, both in the serum exosomes of HCC patients and NT donors. In addition, miRNA is characterized by the length of 18–22 nt (Figure 2). Therefore, serum-exosome-derived miRNAs were the main components of sRNAs and might serve as the key biomarkers for clinical applications.

### 3.2. Three Serum-Exosome-Derived miRNAs Acted as Accurate Biomarkers for the Diagnosis of HCC

Since miRNAs were the most important components in serum exosomes, we further explore the value of serum-exosome-derived miRNAs in the diagnosis of HCC. We firstly detected the base distribution. No significant base distribution bias was observed in the serum-exosome-derived miRNAs between the HCC and non-tumor donors (Appendix A). This represented that the serum-exosome-derived miRNA species had no specificity in HCC. Then, we profiled the serum exosome-derived miRNAs (Appendix A) and detected the differences in the expression levels of specific miRNAs. At the threshold criteria set as log_2_ fold change > 1 and FDR < 0.05, we screened 30 differentially expressed miRNAs (11 miRNAs were upregulated and 19 miRNAs were downregulated) (Appendix A). Therefore, we successfully identified the differences in exosome-derived miRNA expression profiles between HCC and non-tumor donors. Interestingly, among the differentially expressed miRNAs, the miRNAs spliced from the 5’ of the precursor miRNAs accounted for the majority. We identified that 72.7% (8/11) of the upregulated miRNAs and 68.4% (13/19) of the downregulated miRNAs were from 5’ of the precursor miRNAs (Appendix A).

Among these differentially expressed miRNAs, we selected three miRNAs (miR-122-5p: 1.2675 × 10^−7^; let-7d-5p: 0.0023; miR-425-5p: 0.0043) with the top three highest statistical significances as study candidates, according to the rank of FDR values (Figure 3A and Appendix A). To further confirm the three potential biomarkers, we purified serum exosomes and extracted sRNAs from 124 HCC patients and 46 non-tumor donors. To increase the reliability of the results, the samples were divided into a training set and a validation set. From the training set (55 HCC patients and 20 non-tumor donors), three miRNAs were all significantly upregulated in HCC-derived serum exosomes (Figure 3B–D). To calculate the values of the three serum-exosome-derived miRNAs in the diagnosis of HCC, we introduced the levels of the three miRNAs into AUROC mode. The AUROC values of miR-122-5p, let-7d-5p, and miR-425-5p for predicting the diagnosis of HCC were 0.808, 0.702, and 0.660, respectively (Figure 3E,F). If we combined the levels of the three miRNAs, the AUROC value was notably up to 0.905 (Figure 3E,F).

From the training set, the diagnostic performance characteristics of the three miRNAs in diagnosing HCC were shown in Appendix A. The results from the validation set (69 HCC patients and 24 non-tumor donors) also showed that miR-122-5p, let-7d-5p, and miR-425-5p increased significantly in the serum exosomes from HCC, compared with those from non-tumor donors (Figure 4A–C). miR-122-5p, let-7d-5p, and miR-425-5p increased 6.35-fold, 3.10-fold, and 2.12-fold, respectively. The AUROC mode also presented that the three miRNAs had high values in HCC diagnosis (Figure 4D,E). The diagnostic performance characteristics from the validation set are shown in Appendix A. The results were consistent with those from the training set. The diagnostic value of the traditional biomarker AFP in our Chinese cohorts was also calculated. The 3-miRNA signature showed a significantly higher diagnostic value, compared with AFP with a cutoff of 100 ng/mL or 400 ng/mL (Appendix A). Interestingly, if we divided the patients according to the AFP, we found that three miRNAs had higher diagnostic values in the cohort with an AFP larger than 100 ng/mL (Appendix A). The results demonstrated that three serum-exosome-derived miRNAs could be novel biomarkers for HCC diagnosis. By detecting the expression of serum-exosome-derived miR-425-5p, miR-122-5p, and let-7d-5p from 1ml serum, HCC might be accurately and rapidly diagnosed. 

### 3.3. Three Serum-Exosome-Derived miRNAs Were Preferred for the Early Diagnosis of HCC

We also attempted to evaluate the value of the three serum-exosome-derived miRNAs for predicting HCC disease progress or tumor burdens. However, the results showed that three serum-exosome-derived miRNAs were not significantly correlated with tumor size, tumor number, macrovascular invasion, T-stage, AJCC-stage, and the AFP value of HCC (Appendix A). This indicated that HCC patients with a low tumor stage or tumor burden could also express the three serum-exosome-derived miRNAs. Thus, three serum-exosome-derived miRNAs had potential value for the early diagnosis of HCC. 

Then, we detected the application of the three serum-exosome-derived miRNAs for the early diagnosis of HCC. First, the expression levels of the three serum-exosome-derived miRNAs in the low stage of HCC were also significantly higher than those in non-tumor donors (Figure 5A–C). The upregulated fold-changes of miR-122-5p, let-7d-5p, and miR-425-5p were 17.51, 13.88, and 2.36 in the serum exosome of HCC with low-stage, respectively. AUROC mode showed that all three serum-exosome-derived miRNAs could accurately diagnose HCC at its early stage. If we combined the three miRNAs, the AUROC value was up to 0.923 (Figure 5D). Interestingly, compared with the values in the patients with high-stage HCC, the AUROC values in the patients with low-stage HCC were also higher (miR-122-5p: 0.843 vs. 0.791; miR-425-5p: 0.709 vs. 0.599; let-7d-5p: 0.767 vs. 0.589; 3-miRNA signature: 0.923 vs. 0.852; Figure 5D,E). The results from the other patient set also showed that the three miRNAs have significant values in diagnosing HCC at the early stage (Appendix A). Therefore, the three serum-exosome-derived miRNAs were also valuable when the patients suffered from early stage HCC. 

### 3.4. Novel HCC Serum-exosome-derived miRNAs’ Identification

When mapping and annotating the sequences of sRNAs, a number of sRNAs failed to be defined, which suggested that new miRNAs might exist and could be used for the diagnosis of HCC. Thus, we performed the first attempt to identify the novel miRNAs in serum exosomes. As a result, 46 novel miRNAs candidates were identified (Appendix A). Novel miR-224, novel miR-286, and novel miR-300 were the three novel miRNAs with the most significant difference in HCC serum exosomes (Appendix A). The structures of the three novel miRNAs are shown in Appendix A. The analysis of base distribution also showed specific performance. In the serum exosome of HCC, the peak of base A distribution appeared in the middle of the miRNA sequence (Appendix A). However, in the serum exosome of non-tumor donors, the distribution of base A at the head and tail of novel miRNAs (5’ and 3’) increased significantly (Appendix A). In addition, a low proportion of base G existed in the novel miRNAs of the HCC serum exosome (Appendix A). The results suggested that the serum-exosome-derived novel miRNAs in HCC significantly differed from those of the non-tumor donors. However, the diagnostic value and the mechanisms of the novel miRNAs in the serum exosomes of HCC need further investigation. 

## 4. Discussion

Liquid biopsy is a new concept compared with tissue biopsy [25]. At present, tumor tissue biopsy remains the gold standard in tumor diagnosis. However, it does have many defects in the clinic. First, patients must suffer invasiveness during traditional tissue biopsy. Second, tissue biopsy has the potential risk of causing needle tract seeding or dissemination of the tumor [26,27]. Third, tumor heterogeneity also reflected that limited samples from tissue biopsy fail to better collect the complete information of the whole tumor. Thus, increasing interest has been provoked during the practice of blood liquid biopsy. Malignant tumors are characterized by an abundant blood supply, unrestricted proliferation, and invasive behavior [28,29]. This indicates that much more information can be transferred into the blood from tumor cells. Exosomes act as cargo for transferring the contents that contribute to the transmission of cellular information, such as sRNAs, lipids, metabolites, and proteins [30,31]. To date, serum exosome-based liquid biopsy has been widely reported for the diagnosis of many types of human cancers. Serum exosomes can be isolated and obtained efficiently, and the information from serum exosomes could reflect the status of cancers in real time. 

It is well known that serum exosomes contain a variety of sRNAs [32,33]. In addition, the contents of exosomes have the potential to be used for the diagnosis of cancer [34,35,36]. Sorop A et al. reviewed that some exosome-derived miRNAs could be used in the diagnosis of HCC [37]. Exosomal tRNA-derived small RNAs (tsRNAs) also could be the novel diagnostic biomarkers for HCC [38]. However, all the above studies could not present the proportion of sRNAs in serum exosomes from HCC, which is crucial for discovering valuable tumor biomarkers. In our study, we successfully found that miRNAs were the absolutely critical component, compared with other types of exosome-derived sRNAs. However, the reason why HCC cells released abundant miRNAs via exosomes remains unexplained. 

Our study aimed to discover the optimal serum-exosome-derived biomarkers for effective diagnosis of HCC. Some reports have confirmed that miR-224, miR-21, miR-210-3p, miR-93 and miR-155 were upregulated in the exosomes of HCC patients. Xue X et al. also discovered three upregulated miRNAs through exosomal miRNA profiling [39]. Our study confirmed that serum-exosome-derived miR-425-5p, miR-122-5p, and let-7d-5p showed high values in HCC diagnosis. In addition, they were preferred for the diagnosis of early-stage HCC. Although the diagnostic values of the three miRNAs in HCC have never been reported before, they do play a role in tumor regulation. LncRNA SNHG7 promoted the malignancy of HCC by regulating miR-122-5p [40]. MiR-425-5p promoted proliferation and migration of HCC by targeting FOXD3 and RNF11 [41,42]. Let-7d-5p facilitated cisplatin chemosensitivity of ovarian cancer by silencing HMGA1 via the p53 signaling pathway [43]. However, in this study, the question of why the early stage HCC patients released the three exosome-derived miRNAs much more was not concluded. It may involve the complex mechanism of tumor exosomes including miRNA synthesis, exosome production, exosome transfer and release. Alternatively, the three miRNAs act as “pivotal materials” in HCC cells, which contributes to HCC development. Thus, the HCC cells with high malignancy reject to release the three miRNAs or the feedback pathways exist in the malignant HCC that negatively control the synthesis of the three miRNAs. This is worth further investigation.

In addition to the currently known miRNAs, we identified the novel miRNAs that exist in exosomes. Many tools have been developed to mine new miRNAs through bioinformatics approaches [44,45]. Based on the miRNA length, miRNA biogenesis, and stem-loop hairpin secondary structure, several sRNA sequences were recognized as novel serum-exosome-derived miRNAs in this study. We also found that the expression of the novel serum-exosome-derived miRNAs had significant differences between HCC and non-tumor donors. the base distribution of the new miRNAs also had particular features. The distribution of base A had significant differences in the novel serum-exosome-derived miRNAs between HCC patients and non-tumor donors. Meanwhile, base G was not affluent in the serum-exosome-derived novel miRNAs in HCC patients. Thus, we have provided a new insight in that the base distribution of novel miRNAs in serum miRNAs could also work for HCC diagnosis.

There were limitations to this study. The specific mechanisms by which the three miRNAs were upregulated in serum exosomes of HCC deserve attention. The novel serum-exosome-derived miRNAs should also be further explored in a follow-up study. In addition, the results should be validated from other races. The specificity of the three biomarkers should be tested in patients with other forms of cancer or liver tumors.

## 5. Conclusions

In summary, this study focused on the value of in-clinic serum exosomes. We uncovered that miRNAs were the most crucial and valuable sRNAs of serum exosomes. Three serum-exosome-derived miRNAs (miR-122-5p, let-7d-5p, and miR-425-5p) could be practical and accurate biomarkers for HCC diagnosis. They also showed greater values for the diagnosis of early-stage HCC. The identified novel serum-exosome-derived miRNAs could also have the potential for clinical applications as biomarkers for HCC diagnosis. Specifically, exploring serum-exosome-derived miRNAs for diagnosing HCC is promising (Figure 1).

## Data Availability

The datasets used and/or analyzed during the current study are available from the corresponding author on reasonable request.

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
