# Peer review of "Serum-Exosome-Derived miRNAs Serve as Promising Biomarkers for HCC Diagnosis"

_cancers, 2022, doi:10.3390/cancers15010205_

Round 1
Reviewer 1 Report
1. the title of the manuscript needs to be changed to reflect the nature of the study accurately; at the moment, it suggests more of a narrative review,
2. the authors present AFP as a marker of HCC; this is true, but only partially, because the authors should discuss the role of this maker in conjunction with EVs in HCC,
3. according to the ISEV terminology, it is better to use the term "small EVs" rather than "exosomes,"
4. more recent literature should be added to sentence L46 (DOI: 10.3390/cells11182913),
5. figure 1 can serve as a graphical abstract after completing details and quality improvement. I don't find a place for it in the introduction,
6. L72 - the sentence is probably broken off, and what is written there makes no sense,
7. what population were the patients described as non-tumor donors? How were they recruited? Did they differ from the patients in age, sex, and other parameters?
8. figure 2 - it is unclear why only four patients were included in this analysis,
9. the authors should suggest by what mechanisms the studied miRs may induce carcinogenesis,
10. a native speaker must check the English language.
Author Response
Dear Revierew 1:
Thank you for your kind work for us. The responses are attached in the file below.

Reviewer 2 Report
This study is interesting with clinical significance. The authors put forward a new point of view for HCC diagnosis. This diagnostic method has strong operability. The followings are comments to the authors.
1. The morphology of exosomes cannot be clearly seen in Figure 1(B). I suggest that the authors use clear images .
2. The text and number cannot be clearly seen in Figure 1(D),I suggest that the authors use high definition images.
3. Please state how many repeat trials/replicates of each method were conducted in the text and Figure legends? Were the results consistent across all replicates?
4. How many repeat trials/replicates were conducted in Figure 2, Were the results consistent across all replicates? Were the results statistical analyzed?
5. Based on the heterogeneity of cancer patients, the sample size need to be expanded to verify the specificity of serum exosome-derived miRNAs
6.These kinds of studies have limitations. Hence, the author should have stated the potential limitations and suggested what could be done the next step in this area of research.
Author Response
Dear Revierew 2:
Thank you for your kind work for us. The responses are attached in the file below.

Round 2
Reviewer 1 Report
The authors have addressed all my concerns. I consider the manuscript acceptable for publication.
Author Response
Dear Reviewer:
Thanks a lot for your work to improve our quality of manuscript.
Best regards.
Reviewer 2 Report
According to Q1 and Q2, I suggest that the authors show more clear images in the final revision.
Author Response
Dear Reviewer:
Thanks for your work to improve our quality of manuscript. The Figure with high quality is uploaded to the academic Editors.
Best regard.